# FGF19–*FGFR4* Signaling in Hepatocellular Carcinoma

**DOI:** 10.3390/cells8060536

**Published:** 2019-06-04

**Authors:** Aroosha Raja, Inkeun Park, Farhan Haq, Sung-Min Ahn

**Affiliations:** 1Department of Biosciences, Comsats University, Islamabad 45550, Pakistan; aroosha.raja@gmail.com; 2Division of Medical Oncology, Department of Internal Medicine, Gachon University Gil Medical Center, Incheon 21565, Korea; ingni79@hanmail.net; 3Department of Genome Medicine and Science, College of Medicine, Gachon University, Incheon 21565, Korea

**Keywords:** prognosis, *FGF19*, *FGFR4*, HCC, inhibitors

## Abstract

Hepatocellular carcinoma (HCC) is the sixth most common type of cancer, with an increasing mortality rate. Aberrant expression of fibroblast growth factor 19–fibroblast growth factor receptor 4 (*FGF19–FGFR4*) is reported to be an oncogenic-driver pathway for HCC patients. Thus, the *FGF19–FGFR4* signaling pathway is a promising target for the treatment of HCC. Several pan-*FGFR* (1–4) and *FGFR4*-specific inhibitors are in different phases of clinical trials. In this review, we summarize the information, recent developments, binding modes, selectivity, and clinical trial phases of different available *FGFR4*/pan-*FGF* inhibitors. We also discuss future perspectives and highlight the points that should be addressed to improve the efficacy of these inhibitors.

## 1. Introduction

Hepatocellular carcinoma (HCC) is the sixth most common type of cancer, with the fourth highest mortality rate [1]. Despite advancements in therapeutic strategies, the response rate and overall survival rate are still low [2]. The most common cause of HCC is liver cirrhosis from any etiology including hepatitis B and hepatitis C infection, excessive alcohol consumption, diabetes mellitus, and non-alcoholic fatty liver disease [3]. Moreover, various molecular pathways are involved in the initiation and progression of HCC [4]. With respect to these pathways, there is evidence demonstrating the role of fibroblast growth factor pathway genes in HCC prognosis [5].

The fibroblast growth factors (*FGFs*) family comprises a large family of growth factors that are found in different multicellular organisms [6]. The FGFs signal through four transmembrane tyrosine kinase fibroblast growth factor receptors (*FGFRs*) namely *FGFR1*, *FGFR2*, *FGFR3*, and *FGFR4* [7]. *FGFs–FGFRs* are involved in regulation of many biological processes such as embryonic development, cell proliferation, differentiation, and tissue repair [8]. *FGF–FGFR* dysregulation is also widely reported in different types of diseases, disorders, and cancers [9]. Notably, aberrant expression of *FGF19*/*FGFR4* contributes to HCC progression [10].

Since sorafenib marked a new era in molecularly targeted therapy in advanced HCC [11], various drugs such as lenvatinib, regorafenib, cabozantinib, nivolumab, and ramucirumab have subsequently demonstrated overall survival benefits for patients [12,13,14,15,16]. However, the treatment outcome of metastatic HCC is still unsatisfactory, with a median overall survival below 15 months [12]. Thus, more effective treatment options for advanced HCC are needed. This can be achieved by a better understanding of the underlying genetic mechanisms involved in HCC. This review aims to provide comprehensive landscape of current information available on the *FGF19*–*FGFR4* pathway. It also discusses recent advancements on *FGF19*–*FGFR4* inhibitors in HCC. The data is obtained by systematic analysis of the literature and by using different text-mining approaches.

## 2. Overview of *FGFR4* and *FGF19*

### 2.1. Structure and Function of FGFR4

Fibroblast growth factor receptor 4 (*FGFR4*) is a protein coding gene and is a member of tyrosine kinase receptors family. The human *FGFR4* gene is located on chromosome 5 and measures 11.41 bp in length [17]. The *FGFR4* protein coded by two full transcripts of *FGFR4* gene consists of ~800 amino acids, with molecular weight of around 95–110 kDa [18]. The structure of *FGFR4* proteins contains three immunoglobin-like domains (D1–D3), a transmembrane domain, and the kinase domain [19]. (Figure 1) Among these immunoglobin-like domains, first two have role in receptor auto-inhibition, while the third domain is involved in specific binding of ligands [20]. The kinase domain (intracellular) is important in activation of downstream pathways [21]. Further, the kinase domain comprises the N-terminal (smaller) and C-terminal (larger) canonical domains [22]. FGF receptors differ from each other in tissue specificity and ligand-binding affinity. However, good identity scores are found between the kinase domains of *FGFR4* and other FGF receptors [22]. The expression of *FGFR4* is highly tissue-specific due to its unique ligand binding affinity [23]. At a functional level, *FGFR4* is predominantly involved in regeneration of muscles, regulation of lipid metabolism, bile acid biosynthesis, cell proliferation, differentiation, glucose uptake, and myogenesis [24]. Of note, it is reported that *FGFR4* is mostly expressed in liver tissue [25].

### 2.2. FGFR4 in Cancer

*FGFR4* exerts a combination of biological effects that contribute to different hallmarks of cancer (Figure 2) [26]. Functional analysis demonstrated induction of both increased local growth and enhanced metastasis by mutated *FGFR4* [27]. Xu et al. described germline mutations in *FGFR4* i.e., glycine to arginine transition at position 388 in the transmembrane domain of *FGFR4* receptor, which results in the formation of *FGFR4* arg388 allele, leading to higher cancer risk [28]. Due to broad ligand binding spectrum of *FGFR4*, it is reportedly involved in multiple tumor types including HCC, breast cancer, colorectal cancer, rhabdomyosarcoma, and lung cancer [29,30,31,32,33].

### 2.3. Structure and Function of FGF19

Out of three endogenous fibroblast growth factors (*FGF19*, *FGF21*, and *FGF23*), *FGF19* binds to *FGFR4* with highest affinity [34]. The human *FGF19* gene is located on chromosome 11q13. In mice, the *FGF15* gene is an orthologue of the human *FGF19* gene [6]. The farnesoid X receptor (FXR) is activated by the secretion of bile acid from the gall bladder to the small intestine, which ultimately stimulates *FGF19* secretion from the ileum [35,36]. The primary roles of *FGF19* are found in bile acid synthesis, gallbladder filling, glycogen synthesis, gluconeogenesis, and protein synthesis [37]. *FGF19* contributes to several hallmarks of cancer (Figure 3). Interestingly, *FGF19* and *FGF21* (endogenous fibroblast growth factors) are also most commonly involved in regulation of different functions occurring in liver [38]. Nicholes et al. demonstrated in transgenic mice that overexpression of *FGF19* is involved in liver dysplasia [39]. In our recent study, amplification of *FGF19* was found to be significantly associated with cirrhosis and also increased the risk of HCC [40]. Similarly, in our other study we used the fluorescence in situ hybridization technique and found the similar oncogenic patterns of *FGF19* in HCC [41]. Copy number amplification of *FGF19* is also highly reported in The Cancer Genome Atlas (TCGA) data [42]. Notably, the role of *FGF19* at expression level is also frequently reported in HCC prognosis [43,44].

### 2.4. Mechanism of FGFR4 Activation

Specific ligand receptor binding spectrum in FGFs lead to autophosphorylation and formation of multiple complex [45]. *FGFR4* is regulated using its co-receptor klotho-beta (*KLB*) (a transmembrane protein) [46]. The involvement of *KLB* co-receptor is reported in hepatocytes and adipose and pancreatic tissues [47]. *FGFR4* and *KLB* are found to be overexpressed in mature hepatocytes [48]. In addition, *KLB* is required for *FGF19*–*FGFR4* complex activation [49] (Figure 4).

*FGFR4* related pathways have predominant involvement in proliferation, differentiation, survival, and migration of cells. (Figure 4) Multiple signaling cascades such as GSK3β/β-catenin, PI3K/AKT, PLCγ/DAG/PKC, and RAS/RAF/MAPK are modulated by *FGFR4* activation [10,50,51] (Figure 5).

*FGFR4* selectively binds *FGF19* ligand [49,52]. *FGF19* is also reported as a functional partner of *FGFR4*, with the highest score in analysis through the STRING (https://string-db.org/) database.

### 2.5. FGF19–FGFR4 Pathway in HCC

*FGF19*/*FGFR4* activation leads to the formation of FGF receptor substrate 2 (*FRS2*) and growth factor receptor-bound protein 2 (*GRB2*) complex, ultimately activating *Ras*–*Raf*–*ERK1*/*2MAPK* and *PI3K*–*Akt* pathways. (Figure 6) These pathways are predominantly involved in tumor proliferation and anti-apoptosis. (Figure 6).

As discussed, frequent studies reported the anomalous expression of *FGF19*–*FGFR4* complex enhances the progression of HCC [31,44]. In a study conducted on mice model, Cui et al. suggested *FGF19* as a potential therapeutic target for the treatment of HCC [53]. *FGFR4* dysregulation and its correlation with *TGF-β1* also suggested *FGFR4* as potential therapeutic target of HCC patients with invasiveness and metastasis [43,54].

## 3. Targeting *FGF19*–*FGFR4* in HCC

*FGF19*/*FGFR4* inhibition is thought to lead to anti-tumor activities [55]. Thus, several FGFR (1–4) inhibitors are under trial for different types of malignancies including HCC [56] (Figure 7).

### 3.1. Pan-FGFR (1–4) Inhibitors

Multiple pan-*FGFR (1–4)* inhibitors are under-development in different phases of clinical trials (Figure 7). **LY2874455** (NCT01212107), **AZD4547** (NCT02038673), **infigratinib** (NCT02160041), and **erdafitinib** (**NCT02365597**) drugs are designed to target pan-FGFRs and are in phase II of development and clinical trials (Table 1).

**LY2874455** is a small molecule inhibitor developed by Eli Lilly [53] (Figure 8). It has shown promising effects against advanced and metastatic cancers such as myelomas, lung, bladder, and gastric cancer [57]. Its highly effective inhibitory action suggests that it can be effective potential drug for HCC in the near future.

**AZD4547** was developed to specifically target pan-*FGFR (1–4)* in solid tumors. However, **AZD4547** showed good efficacy against *FGFR (1–3)* but weaker activity against *FGFR4* [58], suggesting low efficacy when specifically targeting *FGFR4*.

**Infigratinib** (BGJ398), which targets *FGFR (1–3)* with high affinity and *FGFR4* with less affinity, was developed by Novartis Pharmaceuticals. It is currently in phase II for tumors with alteration of *FGFR* and for glioblastomas, solid tumors, hematologic malignancies, and advanced cholangiocarcinoma. **Infigratinib** showed an effective response against *FGFR* signaling pathways in HCC [59]. However, FDA-approved clinical trials are yet to be conducted for infigratinib in HCC [59].

Janssen Pharmaceuticals reported **erdafitinib** (JNJ-42756493), a pan-*FGFR (1–4)* inhibitor (Figure 9), which is currently under phase II of clinical trials for advanced HCC. It significantly inhibited *FGFR*-overexpressing tumor cells in HCC [60].

**PRN1371** (NCT02608125) and **ASP5878** (NCT02038673) are drugs designed to target pan-*FGFRs* and are in phase I of development and clinical trials. PRN1371 was developed by Principia Biopharma Inc. for solid tumors. It is an irreversible inhibitor that specifically targets *FGFRs*. The inhibitory action of this drug has been reported in many tumor types like HCC, gastric, and lung cancer [61]. Astellas developed **ASP5878** to target pan-*FGFRs (1–4)* in solid tumors. Importantly, ASP5878 also inhibited HCC cell lines exhibiting overexpression of *FGF19* in the pre-clinical phase. In addition, this small inhibitor molecule improved the efficacy of sorafenib [62].

### 3.2. FGFR4-Specific Inhibitors

As discussed, the overexpression of *FGFR4* is most frequently reported receptor compared to FGFR (1–3) in HCC initiation and progression. However, selectivity of pan-*FGFR* inhibitors is comparatively lower for *FGFR4*. Thus, Prieto-Dominguez et al. outlined different targeted therapeutics available for the *FGF19*–*FGFR4* complex [29]. A number of drugs are under different phases of clinical trials which specifically target *FGF19*/*FGFR4*. Two potential drug candidates in the phase II stage of clinical trials, namely **IONIS-*FGFR4*Rx** (NCT02476019) and **FGF-401** (NCT02325739), are reported (Table 2).

**IONIS-*FGFR4*Rx**, previously known as ISIS-*FGFR4*Rx, exhibited antisense inhibitor activity against *FGFR4* [59]. IONIS-*FGFR4*Rx has undergone a phase II clinical trial for obesity, specifically targeting *FGFR4* in liver and fat tissues. It is not only effective in reducing obesity but also improves insulin sensitivity [63]. Thus, we suggest that conducting trials with IONIS-*FGFR4*Rx in HCC patients may give significant results.

**FGF401** was developed by Novartis and specifically targets *FGFR4* in HCC patients. According to the most recent update, FGF401 is in phase II of clinical trials for HCC, expected to be completed by the year 2020. FGF401, with an IC_50_, exhibited at least 1000-fold potency for inhibiting *FGFR4* kinase activity compared to other *FGFRs (1–3)* [64].

**H3B-6527** (NCT02834780), **U3-1784** (NCT02690350), and **BLU-554** (NCT02508467) are reported to be in phase I clinical trials to specifically target *FGFR4*.

**H3B-6527** is a small inhibitor molecule developed by H3 Biomedicine Inc for targeting *FGFR4*-overexpression in advanced HCC and cholangiocarcinoma (IHCC) patients. In preclinical trials, **H3B-6527** proved to be effective in terms of repressing tumor growth in a xenograft model of HCC which exhibited activated aberrant *FGF19*–*FGFR4* signaling [65].

The human monoclonal drug **U3-1784** is under-development by Daiichi Sankyo Inc for HCC and other solid tumors. This antibody specifically binds to *FGFR4* and is most effective (approximately 90%) in *FGF19*-expressing models, suggesting it as a potential drug for HCC with an activated *FGF19*–*FGFR4* pathway. However, according to a recent update, the clinical trials for this drug have been terminated [66].

**BLU-554**, a *FGFR4*-specific inhibitor, is under recruiting phase by Blueprint Medicines Corp. for HCC and cholangiocarcinoma patients. In addition, it was also granted an orphan drug designation in 2015 by the U.S. FDA for HCC [67].

Lastly, **AZ709** showed good selective inhibition of *FGFR4* in HCC, as recently reported by AstraZeneca, and is in the preclinical stage of development. However, no progress has been reported on this drug to date (reported at the 2013 NCRI Cancer Conference, Liverpool, UK).

### 3.3. Irreversible FGFR4 Inhibitors

Two irreversible *FGFR4* inhibitors have also been recently reported, including **INCB62079** (ClinicalTrials.gov Identifier: NCT03144661) and **BLU9931** [68] (Figure 7, Table 3). **INCB62079**, developed by the Incyte Corporation, showed effective dose-dependent and compound-selective activity against cancer cells exhibiting active *FGF19*–*FGFR4*. Additionally, it showed good efficacy in Hep3b hepatocellular cancer xenograft model in pre-clinical trial phase. **INCB62079** is currently in phase I clinical trials (ClinicalTrials.gov Identifier: NCT03144661) for HCC.

Blueprint Medicines Corp reported the remarkable drug **BLU9931**, a small irreversible inhibitor of *FGFR4*. It is currently in the pre-clinical stage of development for HCC and has not been approved by the U.S. FDA. In the preclinical trial phase, BLU9931 exhibited potent antitumor activity in mice with an HCC tumor xenograft with amplified *FGF19* and high expression of *FGF19* at the mRNA level. Recently, it has been reported that *FGF19* shows resistance to sorafenib, but BLU9931 is involved in improving sorafenib efficacy by inactivating *FGFR4* signaling [68].

Apart from the drugs reported in different clinical trials, different studies are underway to find new potent inhibitors against *FGF19*/FGFR. For instance, Cheuk et al. developed a chimeric antibody **3A11ScFvFc** (mice antibody Fv + Human IgG1Fc) to specifically target *FGFR4* in HCC [69]. Chen et al. found ***ABSK-011*** to be involved in suppressing high *FGFR4* expression, which ultimately results in HCC tumor suppression. *ABSK-011*, acting as irreversible inhibitor, selectively modifies *cys552*, which is the residue present within the active site of *FGFR4*. Of note, safety studies have also been conducted for this inhibitor [70]. Lee et al. examined the effect of the **HM81422** inhibitor on the *FGFR4*–*FGFR19* pathway. They successfully demonstrated that **HM81422** can potentially target *FGFR4* activated pathways. However, further elucidation is still required to understand the role of this inhibitor in HCC [71]. Furthermore, different pharmacological approaches suggested significant involvement of the drug sorafenib in inhibiting tyrosine kinase pathways. Initially, Gao et al. reported **sorafenib** as potential tyrosine kinase inhibitor which improves overall survival rate in HCC patients [68]. Later, Matsuki et al. revealed that sorafenib has no particular effect on the oncogenic FGF signaling pathway. However, the involvement of the drug **lenvatinib** was also recently reported [68]. Lenvatinib reportedly inhibits *FGF* pathways in HCC cell lines. Of note, studies suggested that it can be used as a pan-*FGFR (1–4)* inhibitor [68]. However, the specificity of lenvatinib against the *FGF19*–*FGFR4* signaling pathway still remains unclear [72].

## 4. Discussion and Conclusions

Compelling evidence supports the involvement of the *FGF19*–*FGFR4* signaling pathway in HCC [43]. Therefore, this pathway is considered to be a promising therapeutic target for the treatment of HCC. Interestingly, a number of different inhibitors and drugs have been reported to target FGF and *FGFR* signaling pathways. Despite promising advancements, it is still challenging to completely address all the underlying perspectives of this pathway. These perspectives, if clearly addressed, can improve the efficacy and potency of drugs available for HCC. The detailed analysis of available data revealed that *FGFR4* is structurally distinct from other *FGF* receptors (1–3) and also exhibits variable inhibition potency towards different available *FGFR* drugs [73]. Perhaps, this distinct characteristic of *FGFR4* should be exploited in depth to develop *FGFR4*-specific inhibitors to improve drug efficacy for HCC. Importantly, the evidence derived from primates suggests that anti-*FGF19* antibody treatment is mostly accompanied with dose-related liver toxicity [74]. Therefore, the likelihood of adverse effects of FGF/FGFR drugs should be properly envisaged to assure best possible and safe outcomes along with reduced dose-dependent side effects.

In addition, the correlation of *FGF19* gene amplification and HCC is reported to be highly significant, and it is consequently thought to act as potential biomarker for HCC [75]. Therefore, copy number gain of *FGF19* and *FGFR4* should be taken into consideration when designing potential inhibitors of these genes and their pathways.

Conceptually, it is shown that the patients having elevated bile acid concentrations and diabetes have a higher risk of developing HCC [44,53]. Therefore, these complications should be taken into account along with the inhibition of *FGF19*–*FGFR4* pathways to avoid potential adverse impacts and minimize safety risks in HCC patients.

Overall, the degree of *FGF*–*FGFR* inhibition in HCC is not satisfactory. This perhaps gives an indication towards elucidating other factors that are simultaneously involved in the *FGF*–*FGFR* signaling pathway. For instance, *KLB* (the co-receptor of *FGFR4*) is reportedly considered as a novel drug candidate as it is mostly found involved in inducing *FGFR4* overexpression and is also found in an elevated state in HCC [46,76]. Thus, in the future klotho-specific inhibitors can be considered to potentially maximize antitumor and therapeutic benefits in HCC by terminating *FGF19*-binding to *FGFR4*. Lastly, developing drugs that act on key SNPs of *FGFR4* i.e., Gly388 to Arg388, may also be clinically relevant.

In conclusion, most of the *FGFR4*-specific inhibitors are in pre-clinical phases. Progression of these potential inhibitors to advance clinical trial phases coupled with comprehensive research and improvements can revolutionize the available therapeutic options for HCC.

## Figures and Tables

**Figure 1 cells-08-00536-f001:**
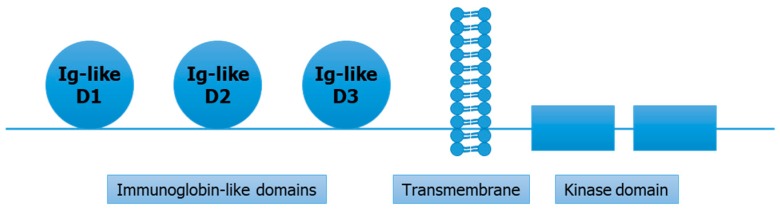
Structural overview of fibroblast growth factor receptor 4 (*FGFR4*) protein.

**Figure 2 cells-08-00536-f002:**
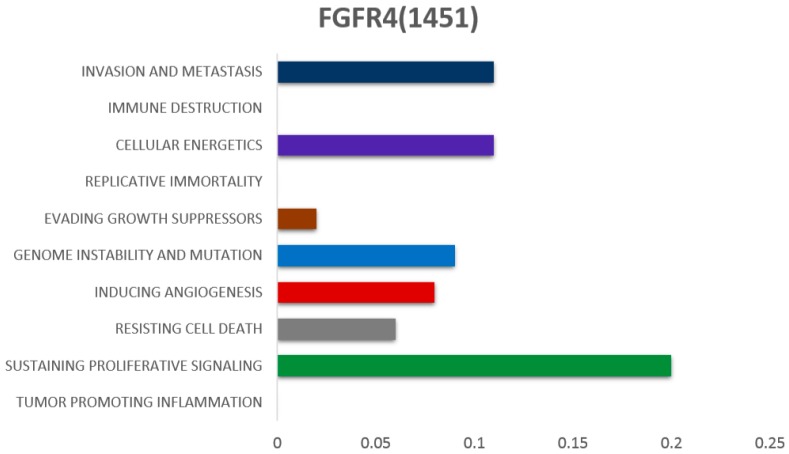
The association of *FGFR4* with different hallmarks of cancer, as reported in the literature. (Scales of bars from left to right represent the lowest to highest number of associations reported)

**Figure 3 cells-08-00536-f003:**
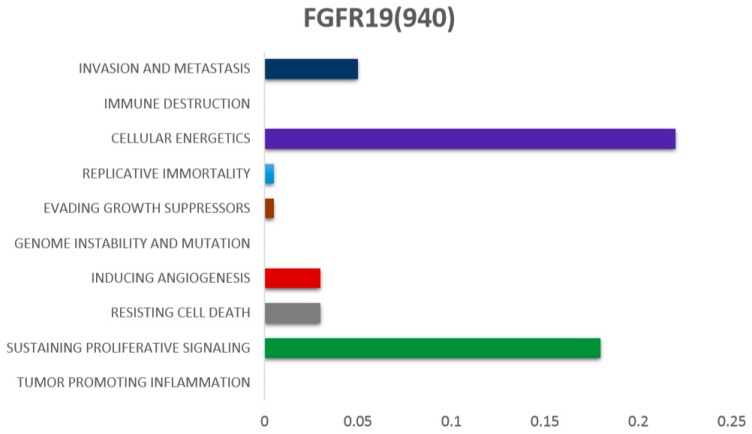
The association of *FGF19* with different hallmarks of cancer, as reported in the literature. (Scales of bars from left to right represent the lowest to highest number of associations reported)

**Figure 4 cells-08-00536-f004:**
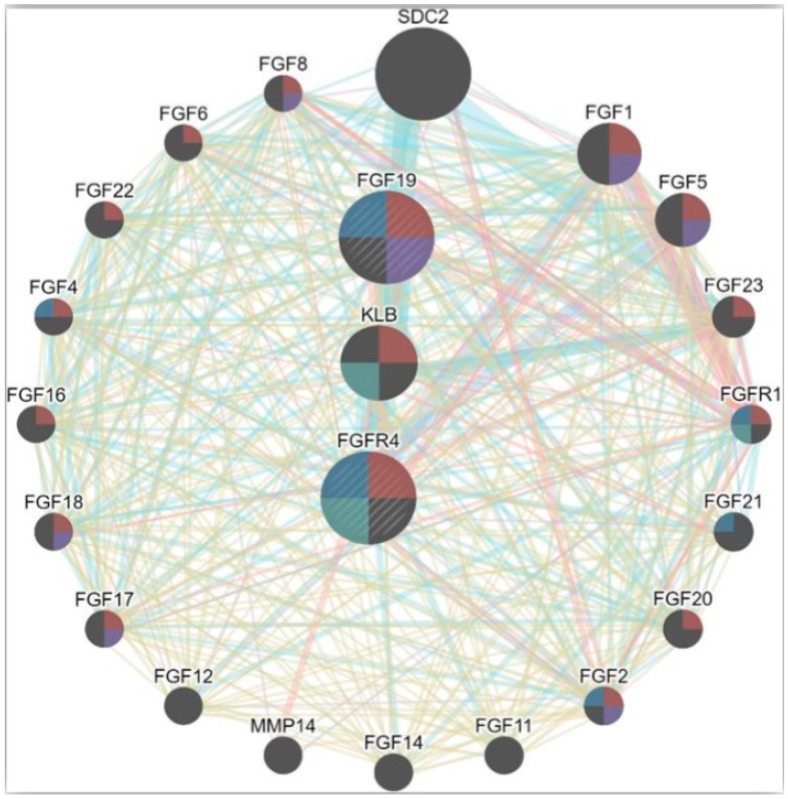
Interaction network of *FGFR4* with different genes with high potency and functional similarity. The interaction network is based on various parameters including co-expression, genetic interactivity, shared protein domains, co-localization and physical interactions.

**Figure 5 cells-08-00536-f005:**
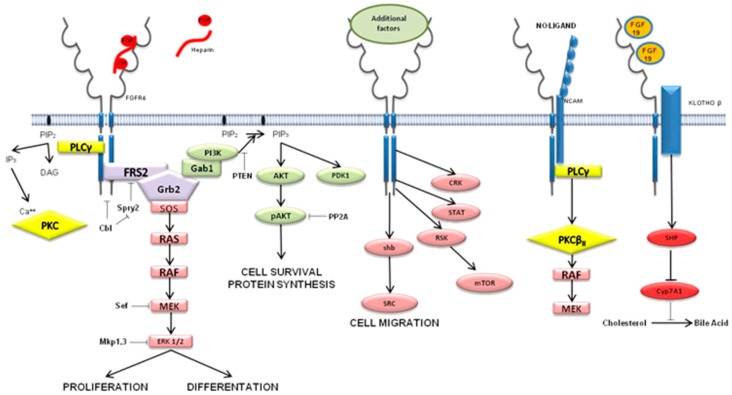
Involvement of *FGFR4*-related signaling pathways. Involvement in cell proliferation is depicted on the far left; next to it the cell survival signaling pathway is shown, and on the right side the cell migration pathway is explained (adapted from Atlas of Genetics and Cytogenetics in Oncology and Haematology).

**Figure 6 cells-08-00536-f006:**
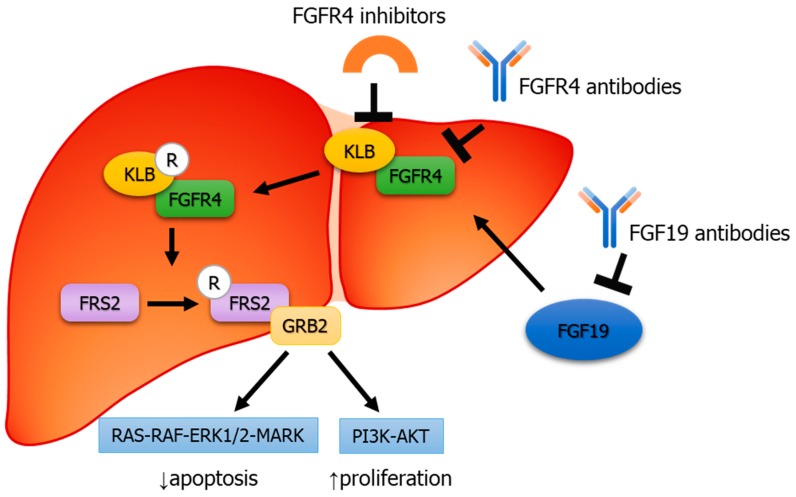
Binding mechanism of *FGF19* to *FGFR4* leads to *FRS2* along with recruitment of growth factor receptor-bound protein 2 (*GRB2*), ultimately leading to activation of the *Ras*–*Raf*–*ERK1*/*2 MAPK* and *PI3K*–*Akt* pathways.

**Figure 7 cells-08-00536-f007:**
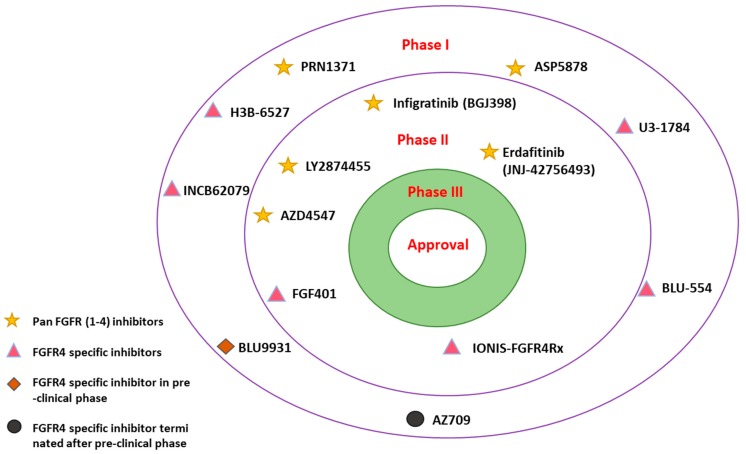
Selected overview of pan-*FGFRs* and *FGFR4*-specific inhibitors in different stages of clinical trials for hepatocellular carcinoma (HCC).

**Figure 8 cells-08-00536-f008:**
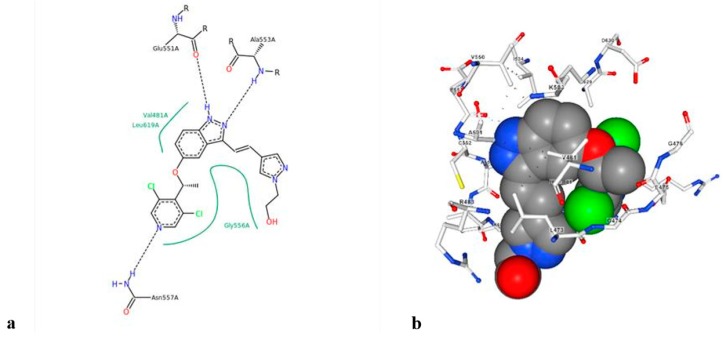
(**a**) Structure of LY2874455, and (**b**) binding mode of LY2874455 with the *FGFR4* kinase domain (PDB code 5JKG).

**Figure 9 cells-08-00536-f009:**
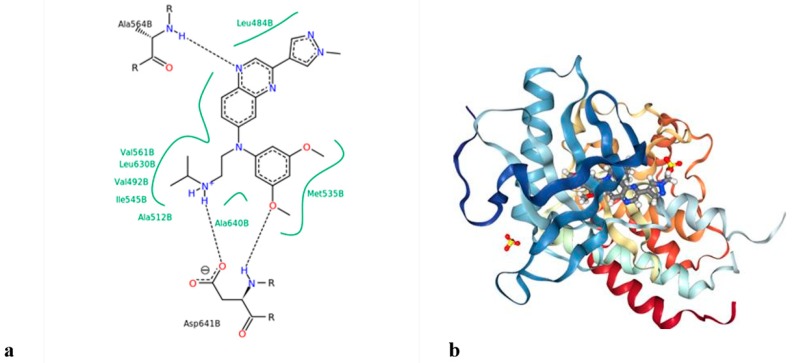
(**a**) Structure of JNJ-42756493 (**b**) Interaction of JNJ-42756493 with FGFR1 (PDB code 5EW8).

**Table 1 cells-08-00536-t001:** Pan-FGFR inhibitors in different phases of clinical trials.

Drug	Company	Indication	Drug Target	Study Phase	Route of Administration	Clinical Trial ID
LY2874455	Eli Lilly	Advanced and metastatic cancers	Pan-FGFR (1–4) inhibitor	Phase II	Oral	NCT01212107
AZD4547	Astra Zeneca	Stage IV squamous cell lung cancer	Pan-FGFR (1–4) inhibitor	Phase II	Oral	NCT02965378
		ER+ breast cancer				NCT01791985
		Muscle-invasive bladder cancer (MIBC)		Phase I		NCT02546661
Infigratinib (BGJ398)	Novartis Pharmaceuticals	Tumors with FGFR genetic alterations	Pan-FGFR (1–4) inhibitor	Phase II	Oral	NCT02160041
		Advanced or metastatic cholangiocarcinoma		Phase II		NCT02150967
		Recurrent resectable or unresectable glioblastoma		Phase II		NCT01975701
		Solid tumor		Phase I		NCT01697605
		Advanced solid malignancies		Phase I		NCT01004224
Erdafitinib (JNJ-42756493)	Janssen Pharmaceuticals	Urothelial cancer Advanced hepatocellular carcinoma	Pan-FGFR (1–4) inhibitor	Phase II	Oral	NCT02365597
		Advanced non-small lung cancer Esophageal cancer				NCT02699606
		Lymphoma				NCT02952573
PRN1371	Prinicipia Biopharma Inc.	Solid tumor	Pan-FGFR (1–4) inhibitor	Phase I	Oral	NCT02608125
ASP5878	Astellas	Solid tumor	Pan-FGFR (1–4) inhibitor	Phase I	Oral	NCT02038673

ER+ breast cancer: estrogen-receptor-positive breast cancer.

**Table 2 cells-08-00536-t002:** *FGFR4*-specific inhibitors under different phases of clinical trials.

Drug	Company	Indication	Drug Target	Study Phase	Route of Administration	Clinical Trial ID
IONIS-*FGFR4*Rx	Ionis Pharmaceuticals	Obesity and insulin sensitivity	*FGFR4*-specific	Phase II	Subcutaneous	NCT02476019
FGF401	Novartis AG	Hepatocellular carcinoma Solid malignancies	*FGFR4*-specific	Phase II (recruiting status)	Oral	NCT02325739
H3B-6527	H3 Biomedicine Inc.	Hepatocellular carcinoma	*FGFR4*-specific	Phase I	Oral	NCT02834780
U3-1784	Daiichi Sankyo Inc.	Advanced solid tumor Hepatocellular carcinoma	*FGFR4*-specific	Phase I (Terminated)	Intravenous	NCT02690350
BLU-554	Blueprint Medicines Corp.	Hepatocellular carcinoma (orphan drug designation for HCC by the U.S. FDA)	*FGFR4*-specific	Phase I	Oral	NCT02508467
AZ709	AstraZeneca	Hepatocellular carcinoma	*FGFR4*-specific	Inactive (Pre-clinical)	Unspecified	

U.S. FDA: U.S. Food and Drug Administration.

**Table 3 cells-08-00536-t003:** *FGFR4*-specific irreversible inhibitors under different phases of clinical trials.

Drug	Company	Indication	Drug Target	Study Phase	Route of Administration	Clinical Trial ID
INCB62079	Incyte Corporation	Liver cancer	*FGFR4*-specific(irreversible)	Phase I	Unspecified	NCT03144661
BLU9931	Blueprint Medicines Corp.	Hepatocellular carcinoma	*FGFR4*-specific(irreversible)	Pre-clinical	Oral

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
