# Peer review of "FGF19–FGFR4 Signaling in Hepatocellular Carcinoma"

_cells, 2019, doi:10.3390/cells8060536_

Round 1
Reviewer 1 Report
The manuscript by Aroosha Raja and colleagues, entitled “FGF19-FGFR4 signaling in hepatocellular carcinoma” is a comprehensive review that collects the information, recent developments, binding modes, selectivity and clinical trial phases of different available FGFR4/pan FGF inhibitors and it is supported by an excellent reference list.
I have just few minor comments that should be considered:
Line 60: Please correct “kD” into “kDa”
Figure 1 appears too simple, please improve it
In Figure 6 please reduce the "T" thickness
In Figure 7 please write the name of pan FGFRs and FGFR4 specific inhibitors in bold or bigger.
Author Response
Reviewer #1
The manuscript by Aroosha Raja and colleagues, entitled “FGF19-FGFR4 signaling in hepatocellular carcinoma” is a comprehensive review that collects the information, recent developments, binding modes, selectivity and clinical trial phases of different available FGFR4/pan FGF inhibitors and it is supported by an excellent reference list.
I have just few minor comments that should be considered:
1. Line 60: Please correct “kD” into “kDa”
Answer)
Thank you for pointing it out. It has been modified in the manuscript.
2. Figure 1 appears too simple, please improve it
Answer)
We have updated Figure 1 to a better version in the manuscript by our best possible efforts. The updated figure clearly illustrates the underlying mechanism.
3. In Figure 6 please reduce the "T" thickness
Answer)
The said suggestion is implemented accordingly. The “T” thickness in reduced from 0.4 to 0.3 inches.
4. In Figure 7 please write the name of pan FGFRs and FGFR4 specific inhibitors in bold or bigger.
Answer)
The figure 7 has been updated as per the suggestion. Bold font is used to present the names of pan FGFRs and FGFR4 specific inhibitors.
Reviewer 2 Report
In this article Raja et al., have performed a review of the current literature on the involvement of FGF19-FGFR4 signaling pathway in the pathogenesis of hepatocellular carcinoma (HCC) as well as current therapeutic strategies that target it. They have also included information on specific inhibitors used in clinical trials.
Overall, this is an interesting work because authors have analyzed all drugs used for blocking FGF19-FGFR4 signaling and they have also included advantages and disadvantages of each one as well as information on the clinical trial stage in which each one was used.
However, the following should be addressed in order for this review to be published:
1) In the abstract, authors mention for the first time FGF19-FGFR4 without giving the full name (i.e Fibroblast growth factor 19- Fibroblast growth factor receptor 4)
2) On page 2, in 2.1 section (line 71), the authors state that «many studies reported…..» however only study is actually mentioned (one reference).
3) In line 104 the authors refer for the first time to klotho-beta and they do not mention the abbreviation in parenthesis (KLB).
4) Figure 4 is blurred. A higher resolution image should be provided. Moreover, the authors may want to consider replacing the specific figure with one that also includes the interaction between KLB and FGFR4
5) The sentence in line 117 « In addition KLB is required… » may be transported in line 106 which is mentioning KLB and thus Figure 5 may be inserted before Figure 4.
6) Since section 2.1-2.5 include an overview of FGFR4 and FGF19, the information on «targeting FGF19-FGFR4 in HCC» that follows could have been presented as a separate section
7) Authors, have to add reference in line 213 which refers to Table 3.
Author Response
Reviewer #2
In this article Raja et al., have performed a review of the current literature on the involvement of FGF19-FGFR4 signaling pathway in the pathogenesis of hepatocellular carcinoma (HCC) as well as current therapeutic strategies that target it. They have also included information on specific inhibitors used in clinical trials.
Overall, this is an interesting work because authors have analyzed all drugs used for blocking FGF19-FGFR4 signaling and they have also included advantages and disadvantages of each one as well as information on the clinical trial stage in which each one was used.
However, the following should be addressed in order for this review to be published:
1) In the abstract, authors mention for the first time FGF19-FGFR4 without giving the full name (i.e Fibroblast growth factor 19- Fibroblast growth factor receptor 4)
Answer)
Thank you for your valuable suggestion. It is updated accordingly in the manuscript.
2) On page 2, in 2.1 section (line 71), the authors state that «many studies reported…..» however only study is actually mentioned (one reference)
Answer)
We agree with the reviewer that there is only one reference mentioned so the said line is updated to “it is reported” instead of many studies reported.
3) In line 104 the authors refer for the first time to klotho-beta and they do not mention the abbreviation in parenthesis (KLB).
Answer)
Thank you for pointing it out. In the updated version, (KLB) is mentioned as an abbreviation while referring to klotho-beta for the first time in manuscript.
4) Figure 4 is blurred. A higher resolution image should be provided. Moreover, the authors may want to consider replacing the specific figure with one that also includes the interaction between KLB and FGFR4.
Answer)
We agree with the reviewer on the issue of image quality. New image of a better resolution is updated in the manuscript. Regarding the suggestion of new image, it unfortunately cannot be changed as it is not only illustrating the interaction between KLB and FGFR4 but is also referring towards the potential role of FGFR4 related pathways in proliferation, differentiation, survival and migration of cells.
5) The sentence in line 117 « In addition KLB is required… » may be transported in line 106 which is mentioning KLB and thus Figure 5 may be inserted before Figure 4.
Answer)
While agreeing with the reviewer on the said point, modifications in manuscript are made accordingly. The Figure 4 is updated to Figure 5 and vice versa. Similarly, the line “In addition KLB is required” is shifted as per the suggestion.
6) Since section 2.1-2.5 include an overview of FGFR4 and FGF19, the information on «targeting FGF19-FGFR4 in HCC» that follows could have been presented as a separate section
Answer)
We are in complete agreement with the viewpoint of the reviewer. “Targeting FGF19-FGFR4 pathway in HCC” is now placed in a separate section i.e. the 3rd section.
7) Authors, have to add reference in line 213 which refers to Table 3.
Answer)
The ClinicalTrials.gov Identifier and reference for the two irreversible inhibitors mentioned in the line are updated in the manuscript. All the suggestions and minor comments are addressed with the best possible efforts.